# LTα, TNF, and ILC3 in Peyer’s Patch Organogenesis

**DOI:** 10.3390/cells11121970

**Published:** 2022-06-19

**Authors:** Violetta S. Gogoleva, Dmitry V. Kuprash, Sergei I. Grivennikov, Alexei V. Tumanov, Andrey A. Kruglov, Sergei A. Nedospasov

**Affiliations:** 1Center for Precision Genome Editing and Genetic Technologies for Biomedicine, Engelhardt Institute of Molecular Biology, Russian Academy of Sciences, 119991 Moscow, Russia; viogogoleva@eimb.ru (V.S.G.); kuprash@eimb.ru (D.V.K.); sergey.grivennikov@cshs.org (S.I.G.); kruglov@drfz.de (A.A.K.); 2Department of Microbiology, Immunology and Molecular Genetics, University of Texas Health Science Center, San Antonio, TX 78229, USA; tumanov@uthscsa.edu; 3Belozersky Institute of Physico-Chemical Biology and Faculty of Biology, Lomonosov Moscow State University, 119234 Moscow, Russia; 4AG Chronic Inflammation, Deutsches Rheuma-Forschungszentrum (DRFZ), A Leibniz Institute, 10117 Berlin, Germany; 5Institute of Cell Biology and Neurobiology, Charite-Universitatsmedizin, 10117 Berlin, Germany; 6Division of Immunobiology and Biomedicine, Sirius University of Science and Technology, 35340 Federal Territory Sirius, Russia

**Keywords:** TNF, lymphotoxin, Peyer’s patches, lymphoid tissue organogenesis, innate lymphoid cells

## Abstract

TNF and LTα are structurally related cytokines of the TNF superfamily. Their genes are located in close proximity to each other and to the *Ltb* gene within the TNF/LT locus inside MHC. Unlike *Ltb*, transcription of *Tnf* and of *Lta* is tightly controlled, with the *Tnf* gene being an immediate early gene that is rapidly induced in response to various inflammatory stimuli. Genes of the TNF/LT locus play a crucial role in lymphoid tissue organogenesis, although some aspects of their specific contribution remain controversial. Here, we present new findings and discuss the distinct contribution of TNF produced by ILC3 cells to Peyer’s patch organogenesis.

## 1. Introduction

Tumor necrosis factor (TNF) and lymphotoxin alpha (LTα) are the first molecularly cloned cytokines of the TNF superfamily with similar cytotoxic activities of their recombinant gene products [1]. Both TNF and LTα can form soluble homotrimers, sTNF and sLTα_3_, respectively, that bind to and signal through both TNFR1 and TNFR2 [2], although some studies question the physiological relevance of LTα_3_-TNFR2 interaction [3,4]. Renaming of lymphotoxin as TNFβ caused some confusion, which was resolved later with the discovery of LTβ [5], uncovering a constellation of LTα functions that are not shared by TNF [6].

Although LTα can be a soluble TNF-like factor in vivo [7], its major forms are membrane-bound heterotrimeric complexes with LTβ, such as LTα_1_β_2_ and LTα_2_β_1_. While LTα_1_β_2_ signals through unique LTβR [8], LTα_2_β1 can interact both with TNFR1 and TNFR2 [9]. A further complication in the TNF/LT ligand–receptor system is the fact that LIGHT, another member of TNF superfamily, being a ligand for HVEM, may also serve as a ligand for LTβR [10].

Since biologics targeting TNF (or both TNF and soluble LTα) are used for treatment of autoimmune diseases, it is of great importance to understand all functions mediated by the ligands of TNF/LT system in normal physiology. As models for such an analysis, multiple engineered mouse strains with complete or conditional inactivation of corresponding genes were generated (reviewed in [11,12]). These studies unexpectedly uncovered that both TNF and LT are involved in the development and the structure-functional organization of secondary lymphoid organs, including lymph nodes and Peyer’s patches (PP) in the small intestine. PP development during gestation relies on the activation of the LTα_1_β_2_/LTβR axis, as PP are completely absent in all independently generated *Lta*-, *Ltb*- and *Ltbr*-deficient mice [13,14,15,16,17]. As it was earlier demonstrated for lymph node development, PP formation requires expression of LTα_1_β_2_ on the lymphoid-tissue inducer (LTi) cell, a specialized type 3 innate lymphoid cell [18]. However, uniquely for PP, LTα_1_β_2_ expression on LTi is induced after the activation of tyrosine kinase receptor RET signaling in CD11c^+^LTα_1_β_2_^+^ haematopoietic cells [19]. LTα_1_β_2_-expressing LTi then activate LTβR^+^ mesenchymal cells, leading to the upregulation of cell adhesion molecules and chemokines, such as CXCL13, CCL19 and CCL21, that attract and retain lymphocytes at the site of the developing PP [20].

Although we have a detailed knowledge of how lymph node development is orchestrated by TNF superfamily molecules, some issues concerning PP development remain unresolved. First, the physiological thresholds of TNF/LTα signals required for proper PP development remain debatable. For example, double heterozygosity of both *Lta* and *Ltb* results in the complete lack of PP, implicating the importance of gene dosage during the process of PP formation [21]. Second, the contribution of TNF to PP development also remains controversial. Several independently generated *Tnfr1*-deficient mouse strains display a range of defects in PP organogenesis [22,23,24]. In line with this, PP formation in several independently designed *Tnf*-deficient mouse strains was either reduced [23,25] or completely absent [26]. To address the role of TNF and LTα in the process of PP formation in this study, we relied on several *neo*-free knockout strains generated using LoxP-Cre technology [14,16,26]. Our findings with these mice do not support earlier conclusions on the threshold requirements for LTα/LTβ signal involvement and they implicate a role for TNF produced by ILC3 in PP development.

## 2. Materials and Methods

### 2.1. Mice

*Lta*^Δ/^^Δ^ [14], *Ltb*^Δ/^^Δ^ [16], *Tnf*^Δ/^^Δ^ [26], *Tnf*^fl/fl^, *Tnf*^ΔLysM^, *Tnf*^ΔT^ [27] mice were described previously. *Tnfrsf1a*^−/−^ (*Tnfr1*^−/−^) were kindly provided by Prof. K. Pfeffer [28]. *Tnf*^ΔILC3,T^ mice were generated by crossing *Tnf^f^*^l/fl^ and *Rorc*(*γt*)-Cre^Tg^ mice [11]. *Tnf*^ΔCD11c^ were generated by crossing *Tnf^f^*^l/fl^ and *CD11c*-Cre mice [29]. Experimental mice were either generated on or extensively backcrossed onto a C57BL/6 genetic background. Heterozygous (*Lta*^+/^^Δ^) and littermate (*Tnf*^fl/fl^) controls were used as wild-type controls. All mice analyzed were sex- and age-matched (8–12 weeks old). Mice were bred and housed under the standard conditions of the Animal Breeding Facility, BIBCh, RAS (the Unique Research Unit Bio-Model of the IBCh, RAS; the Bioresource Collection—Collection of SPF-Laboratory Rodents for Fundamental, Biomedical and Pharmacological Studies under contract #075-15-2021-1067 with the Ministry of Science and Higher Education of the Russian Federation) accredited at the international level by AAALAC. All animal experiments were approved by the local authorities IACUC committee of BIBCh RAS (protocol no. 125, 29 December 2020) and performed according to institutional guidelines and Bioethics Committee of the Engelhardt Institute of Molecular Biology, Russian Academy of Sciences.

### 2.2. Visualization of Peyer’s Patches

The small intestine was removed and placed in 10% acetic acid for 5–10 min, then Peyer’s patches were counted. Images of intestines were taken with a consumer digital camera. Whole-mount immunostaining was carried out as described in [26]. For staining of the PP anlagen, rat anti-MAdCAM-1 mAb MECA-367 (BD Biosciences, San Jose, CA, USA) was used with horseradish peroxidase conjugated anti-rat Ig (Jackson ImmunoResearch, West Grove, PA, USA) as a secondary antibody, and SIGMA FAST DAB with a metal enhancer (Sigma-Aldrich, St. Louis, MO, USA) was used for the enzymatic reaction.

### 2.3. Cell Isolation from the Intestinal Lamina Propria

The small intestines were extracted and placed into the ice-cold PBS. Peyer’s patches and fat were removed. The small intestine was cut longitudinally and cleaned of the feces. Subsequently, the tissue was twice incubated in HBSS supplemented with 10 mM HEPES and 5 mM EDTA at 37 °C for 20 min. Samples were twice washed, minced, and incubated in HBSS supplemented with 2% FBS, 0.5 mg/mL Collagenase D (Roche, Mannheim, Germany), 0.5 mg/mL DNase I (Sigma Aldrich), 0.5 U/mL Dispase (Gibco, Waltham, MA, USA) at 37 °C for 20 min. After filtering through a 70 μm strainer, cells were centrifuged in a 40/80 Percoll gradient [30].

### 2.4. Flow Cytometry and Cell Sorting

Cells were stained with Fixable Viability Dye (eBioscience, San Diego, CA, USA) and the following antibodies: anti-CD45 (30-F11, BioLegend, London, UK), anti-CD4 (RM4-5, BioLegend), anti-CCR6 (29-2L17, BioLegend), anti-IL-7Rα (A7R34, Invitrogen, Waltham, MA, USA), Mouse Hematopoietic Lineage Cocktail (eBioscience): anti-CD3 (17A2), anti-B220 (RA3-6B2), anti-CD11b (M1/70), anti-TER-119 (TER-119), anti-Gr-1 (RB6-8C5). Intracellular staining was performed using an eBioscience Foxp3/Transcription Factor Staining Buffer Set and the following antibodies: anti-RORγt (B2D, eBioscience), anti-GATA3 (16E10A23, BioLegend). Flow cytometric analysis was performed using FACSAria III flow cytometer (BD Bioscience) and FlowJo software (Tree Star). Cell sorting of splenic ILC3 and CD4^+^ T cells was performed using anti-CD4 (GK1.5; eBioscience) and Hematopoietic Lineage Cocktail (anti-CD3, anti-CD19, anti-CD11c) on FACSAria (BD Bioscience). ILC3 were identified as Lineage^−^CD4^+^ cells, CD4^+^ T cells were identified as Lineage^+^CD4^+^ cells. Purity was more than 90%.

### 2.5. Quantitative PCR Analysis

Total RNA was isolated using TRI reagent (Sigma-Aldrich), according to the manufacturer’s protocol. RNA was treated with DNase I (Thermo Fischer Scientific, Waltham, MA, USA; Cat #: EN0525), according to the manufacturer’s instructions. To transcribe RNA to cDNA, random hexamer primers (TIB Molbiol, Berlin, Germany) and RevertAid H Minus reverse transcriptase (Thermo Fischer Scientific; Cat #: EP0451) were used, according to the manufacturer’s protocol. The generated single-stranded cDNA was directly used for quantitative real-time PCR. cDNA transcript expression of indicated genes was analyzed with Maxima SYBR Green/ROX qPCR master mix (Thermo Fischer Scientific; Cat #: K0222), according to the manufacturer’s protocol by using a StepOne Plus Real Time PCR system (Applied Biosystems, Foster City, CA, USA) or Mx3000P qPCR system (Stratagene, La Jolla, CA, USA).

### 2.6. Statistical Analysis

All statistical analyses were performed using Prism 9 software (GraphPad, San Diego, CA, USA). Unpaired *t* test and one-way ANOVA tests were used. *p* values of <0.05 were considered significant.

## 3. Results

### 3.1. Physiological Thresholds for LTα and TNF Signaling in Peyer’s Patch Development

The differential contribution of LTα and LTβ to the development of mesenteric lymph nodes implicated distinct physiological thresholds for TNF- and lymphotoxin-dependent activation of mesenchymal cells. Interestingly, Koni et al. reported that unlike single heterozygotes, the double heterozygous *Lta*^+/−^*Ltb*^+/−^ mice completely lacked PP [21]. However, this phenotype may be explained by the possible cooperation of TNFR1 and LTβR downstream pathways in PP formation [21]. Additionally, both *Lta*^−/−^ [13] and *Ltb*^−/−^ [15] knockout mice retained the *neo* cassette that could possibly affect the expression of neighboring genes, for example, *Tnf* (Appendix A). Importantly, the presence of the *neo* cassette may affect the expression of neighboring genes [31,32]. Therefore, we reexamined the effects of the LTα/LTβ gene dosage in PP organogenesis using *neo*-free *Lta*^Δ/^^Δ^ and *Ltb*^Δ/^^Δ^ mice [14,16] on a pure C57BL/6 background (Appendix A). Surprisingly, these “alternative” *Lta*^+/^^Δ^/*Ltb*^+/^^Δ^ double heterozygotes developed PP normally, as did single heterozygous *Lta*^+/^^Δ^ (Figure 1A) and *Ltbr*^+/−^ mice (data not shown).

We then crossed the conventional *Lta*^−/−^ mice [13] with *neo*-free *Ltb*^Δ/^^Δ^ mice and observed an intermediate phenotype with a significantly reduced numbers of PP (Figure 1A). As reported earlier, *Lta*^−/−^ mice, unlike *Lta*^Δ/^^Δ^ mice, showed a diminished ability of myeloid cells to produce TNF due to the interference of the *neo* cassette with the far upstream elements of TNF promoter/enhancer region [14,33]. These results suggest that impaired TNF production is responsible for defective PP development in LTα haploinsufficient mice. Of note, the PP development was not affected in conventional single heterozygous *Lta*^+/−^ mice [21], suggesting that *Ltb* gene dosage may also be important at certain gestational steps in PP formation.

### 3.2. TNF Produced by ILC3 Is Critical for Peyer’s Patches Development

Since it is known that soluble LTα_3_ and TNF share the TNFR1 but not the LTβR receptor, we next re-addressed the requirement for TNF and TNFR1 during PP development. As reported earlier, neither *neo*-free *Tnf*^Δ/^^Δ^ (Appendix A) nor *Tnfr1*-deficient mice— both on a C57BL/6 background—developed PP (Figure 1B), excluding the effects of genetic modifiers affecting the phenotype [28].

Analysis of *Tnf*^Δ/^^Δ^ E16.5 embryos showed that they lacked PP anlagen [26]. This finding indicates that TNF may be required during the early stages of PP development, i.e., at the induction phase of LTα_1_β_2_ expression or clustering of LTβR^+^ mesenchymal cells and LTi cells. As RET^+^CD11c^+^ hematopoietic progenitor cells contribute to the early steps of PP formation [18] we therefore analyzed PP development in mice with the inactivation of TNF in LysM^+^ cells [34] and CD11c^+^ cells (Figure 2A).

We found that TNF derived from LysM^+^ myeloid cells and CD11c^+^ cells is dispensable to PP formation (Figure 2A).

To define the contribution of ILC3-derived TNF, we generated mice with conditional inactivation of TNF in ILC3 and T cells by crossing *Tnf^f^*^l/fl^ and *Rorc*(*γt*)-Cre^Tg^ mice [11]. Since ILC3 cells are known to express high levels of LTα, LTβ, and TNF mRNA [35], we first analyzed the expression of TNF/LT locus genes in purified ILC3 and CD4^+^ T cells isolated from the spleens of adult *Tnf*^ΔILC3,T^ and *Tnf*^fl/fl^ mice. We found no difference in LTα and LTβ mRNA levels between wild-type and *Tnf*^ΔILC3,T^ mice (Figure 2B) and no detectable TNF mRNA in either ILC3 or CD4^+^ T cells (Figure 2B). Importantly, *Tnf*^ΔILC3,T^ mice completely lacked PP, whereas *Tnf*^ΔT^ displayed normal PP numbers (Figure 2A). These results reveal that ILC3-derived TNF is critical for PP organogenesis. To sum up, these results suggest that activation of the TNF/TNFR1 axis is important for PP formation, with the specific contribution of ILC3-derived TNF.

### 3.3. ILC3 in PP Development and in Adult Lamina Propria

Peyer’s patches represent secondary lymphoid organs in the small intestine implicated in both the induction of immune tolerance and in defense against certain microorganisms [36]. Although 8 week old *Tnf*^Δ/^^Δ^ and *Tnf*^ΔILC3,T^ mice showed a complete absence of PP, they both developed a single lymphoid patch at the cecum of the large intestine (Figure 3A).

These results indicate that TNF is dispensable for cecal patch development. Further immunohistochemical staining for MAdCAM-1 revealed the absence of PP anlagen in newborn *Tnf*^Δ/^^Δ^ and *Tnf*^ΔILC3,T^ mice (Figure 3A).

Since ILC3 subsets are involved in both lymphoid organ formation and in regulation of gut immune homeostasis, we next analyzed ILC populations in the lamina propria of adult TNF/LT deficient animals (Appendix A). We found that the frequency of RORγt^+^ ILC3 in small intestine of *Lta*-deficient mice increased (Figure 3B). Similarly, the frequency of RORγt^+^ ILC3 in mice with either *Tnf* or *Tnfr1* deficiency increased compared to the *Tnf*^fl/fl^ control mice (Figure 3C). Thus, we suggest that the absence of PP is not due to reduced numbers of ILC3. Moreover, these results indicate that TNF/LT ligands not only regulate PP development during embryogenesis but also control the homeostasis of ILC3 in the intestines of adult animals.

## 4. Discussion

TNF and LTα are structurally and functionally related cytokines of the TNF superfamily. They mediate multiple functions, including organogenesis and structural maintenance of lymphoid tissue [37,38]. However, the requirements of TNF and LTα during development may vary for different secondary lymphoid organs [39]. For example, lymph node development (except for mesenteric and sacral LNs) strictly depends on the activation of LTα_1_β_2_ signaling [18], while TNF seems to be dispensable in this process, as lymph nodes are present in mice with *Tnf* [23,26], *Tnfr1* [22,23], or *Tnfr2* deletion [40]. However, TNF clearly contributes to the functional microarchitecture of the spleen and the lymph nodes [41,42].

The significance of functional allele numbers, a so-called gene dosage effect, in genetically modified mice was demonstrated for both TNF and LTα. For example, mice heterozygous for both *Tnf* and *Lta* deficiency produced decreased plasma TNF levels in response to LPS stimulation as compared to wild-type mice [43]. Our demonstration of normal PP development in *Lta*^+/^^Δ^/*Ltb*^+/^^Δ^ double heterozygous mice challenges the interpretation of Koni et al. [21] and calls for an alternative explanation of that apparent paradox. First of all, the requirements of TNF and TNFR1 in PP organogenesis remain controversial, partly due to the close linkage of *Tnf* with *Lta* and *Ltb* and of *Tnfr1* with *Ltbr*. Possibly, the differences in targeting designs for *Tnf*- and *Tnfr1*-deficient mice led to inconsistent phenotypes, particularly concerning PP development. In the current study, we confirmed the complete absence of PP in *neo*-free *Tnf*^Δ/^^Δ^ and *Tnfr1*-deficient mice on a C57BL/6 background. We speculate that the presence of the *neo* cassette in conventional *Tnf*^−/−^ mice and/or the possible contribution of genetic modifiers in *Tnfr1*^−/−^ mice generated on a mixed 129Sv x C57BL/6 background with reduced PP number [23] may have caused a compensatory upregulation of lymphotoxin genes.

With regard to the cellular source of TNF, we identified RORγt^+^ ILC3 but not CD11c^+^ cells as the critical contributor to PP development. ILC3-derived TNF specifically regulates formation of PP in the small intestine but not of the cecal patch in the large intestine. In line with our study, previous work demonstrated that the mechanism of development and maintenance of the cecal patch is distinct from PP and colonic solitary intestinal lymphoid tissues [44,45].

Moreover, ILC3-derived TNF is critical for the gestational development of PP anlagen. Interestingly, in the adult organism TNF clearly contributes to the maintenance of PP. Specifically, B cell-derived TNF controls the development of follicular dendritic cells and B-cell follicles in PP [42,46]. Based on our data, it appears that TNF and LTα regulate the frequency of ILC3 in the intestinal lamina propria. We also do not exclude a possibility that increased numbers of ILC3 in the intestines of PP-deficient *neo*-free *Tnf*^Δ/^^Δ^ and *Lta*^Δ/^^Δ^ mice is a result of compensatory feedback loop mechanism.

As a result, we propose an updated model of a stepwise contribution of TNF and LTα in PP development (Figure 4).

The **first** step (E15.5) is the appearance of VCAM-1^+^ stromal cells clusters, also known as lymphoid tissue organizer cells (LTo) [47]. This step starts from the interaction of VCAM-1^+^ stromal cells producing artemin (ARTN) with RET^+^c-Kit^+^CD11c^+^CD4^−^CD3^−^IL-7Ra^−^ hematopoietic progenitor cells, also known as lymphoid tissue initiator cells (LTin) [19,48]. Initial adhesion and clustering of these cells is dependent on ART-RET interactions and is LTi-independent [48]. These CD11c^+^ hematopoietic progenitors start to express LTα_1_β_2_ at higher levels than the LTi cells at E15.5, possibly due to the activation of RET signaling [19]. However, RET stimulation of explanted cultures from intestines with ART and other RET ligands failed to induce LTβ expression by LTin cells [48]. It is possible that increased LTα_1_β_2_ levels by LTin cells could be due to LTα upregulation, as transcription of *LTA* gene in various cell types is tightly controlled as compared to *LT*β. This is consistent with data regarding the functional activity of an LTα_1_β_2_ complex and that the LTα subunit is a limiting component for the assembly of a functional heteromeric LTα_1_β_2_ complex [49]. LTα_1_β_2_ on CD11c^+^ cells then enables their interaction with LTβR on stromal cells suggesting that this may be a first checkpoint dependent on TNF superfamily ligands signaling.

At the **second** step (E17.5), more LTi cells are recruited to the PP anlagen. Stromal cells produce IL-7 which promotes the survival of LTα_1_β_2_-expressing LTi cells, leading to their interaction with LTβR^+^ stromal cells [50,51]. Here, we propose that in addition to LTα_1_β_2_, LTi cells may produce TNF, because deletion of *Tnf* in ILC3 results in a complete block of PP development. Interestingly, membrane-bound TNF on LTi cells appears to be involved, as mice expressing only this form of TNF develop PP with relatively normal microarchitecture [52]. Although we cannot draw a definite conclusion as to which step of PP development is controlled by TNF, data of *Tnf*^Δ/^^Δ^ embryos that lacked PP anlagen at E16.5 [26] suggest that TNF may be critical during the first two steps of PP formation. Hypothetically, LTin and LTi cells may produce TNF and sLTα_3_ that interact with TNFR1 on an unknown cell, since *Tnfr1*- but not *Tnfr2*-deficient mice lack PP. Recent analysis of mice with conditional inactivation of TNFR1 in CCL19-Cre and Col6A1-Cre expressing cells revealed reduced B cell follicles and the impaired stromal microarchitecture of PP [53]. Additionally, single cell RNA seq analysis and fate mapping experiments identified two distinct fibroblastic reticular cell lineages in PP anlagen at E18.5: perivascular and subepithelial, which synergistically direct PP development in both LTβR- and TNFR1-dependent manner [53].

Starting from E17.5, the **third** step includes production of chemokines attracting hematopoietic cells to the PP, such as CXCL13, CCL19, and CCL21 [18]. These chemokines are produced by stromal cells via activation of non-canonical NF-κB [54]. Engagement of TNFR1 by TNF or LTα is known to activate the canonical NF-κB pathway, leading to the transcription of different target genes, including adhesion molecules [37,54]. Although the role of the non-canonical NF-κB pathway in the development of PP is well established [55,56], mice deficient in p50, one of the canonical NF-κB subunits, possess a significantly reduced number of PP [55]. Moreover, simultaneous deletion of TNFR1 and another NF-κB subunit, RelA, results in the absence of PP, which is explained by the dependence of stromal cells on RelA [57]. We propose that TNFR1-dependent activation of canonical NF-κB is not only crucial for the activation of various proinflammatory genes but also for some homeostatic functions, including the late steps of PP formation. The possible contribution of LTα_3_/TNFR1 to PP development should also be considered since its specific role in the organization of mucosal lymph nodes was demonstrated in double *Ltb*/*Tnf*-deficient mice [41]. This model, with distinct contributions of TNF and LTα molecules into PP formation, resembles the sequence of events described for mature isolated lymphoid follicles [58].

## Figures and Tables

**Figure 1 cells-11-01970-f001:**
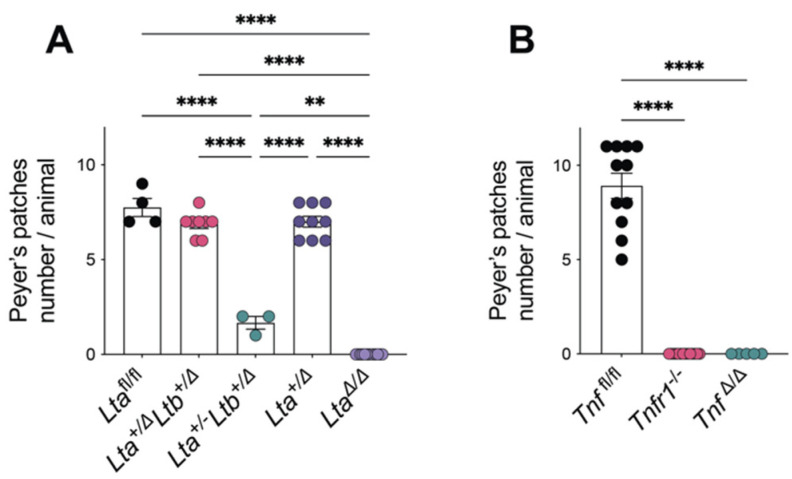
Distinct contribution of various genetic deficiencies involving TNF/LT ligand–receptor system to Peyer’s patch development. (**A**) Peyer’s patches numbers in adult *Lta*^fl/fl^ (*n* = 4), doubly heterozygous *Lta*^+/^^Δ^*Ltb*^+/^^Δ^ (*n* = 8), doubly heterozygous *Lta*^+/−^*Ltb*^+/^^Δ^ (*n* = 3), single heterozygous *Lta*^+/^^Δ^ (*n* = 9), and homozygous *Lta*^Δ/^^Δ^ mice (*n* = 13); (**B**) Peyer’s patches numbers in adult *Tnf*^fl/fl^ (*n* = 11), *Tnfr1*^−/−^ (*n* = 14), and *Tnf*^Δ/^^Δ^ (*n* = 5) mice. Each symbol represents an individual mouse; mean ± SEM. ** *p* < 0.01, **** *p* < 0.0001 (one-way ANOVA).

**Figure 2 cells-11-01970-f002:**
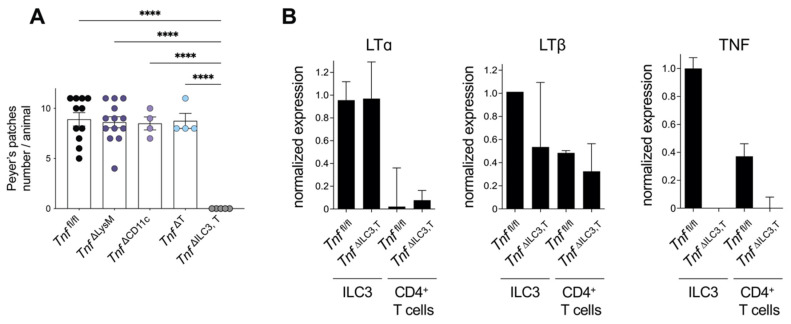
ILC3-derived TNF is critical for Peyer’s patches development. (**A**) Peyer’s patches numbers in adult *Tnf*^fl/fl^ (*n* = 11), *Tnf*^ΔLysM^ (*n* = 13), *Tnf*^ΔCD11c^ (*n* = 4), *Tnf*^ΔT^ (*n* = 4), and *Tnf*^ΔILC3,T^ (*n* = 5) mice; (**B**) mRNA expression of LTα, LTβ, and TNF in purified ILC3 and CD4^+^ T cells isolated from the spleens of *Tnf*^fl/fl^ and *Tnf*^ΔILC3,T^ mice. Each symbol represents an individual mouse; mean ± SEM. **** *p* < 0.0001 (one-way ANOVA).

**Figure 3 cells-11-01970-f003:**
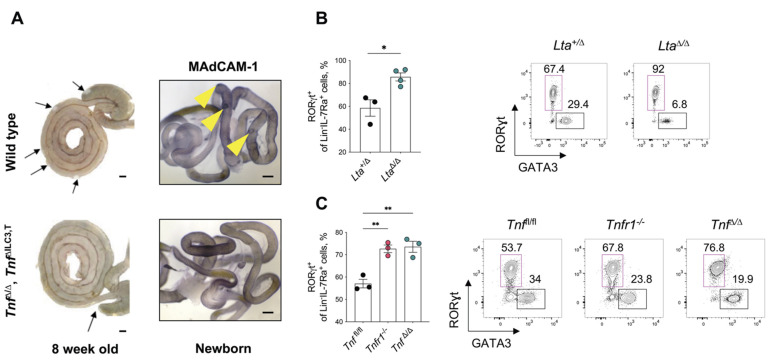
LTα and TNF regulate ILC3 frequency in lamina propria in an adult organism. (**A**) Photographs of mouse intestine, from representative animals, showing the absence of Peyer’s patches and the presence of a cecal patch (**left**) and immunohistochemical staining of intestine from newborn mice (**right**). Arrows indicate Peyer’s patches (**left**) and areas of specific MAdCAM-1 staining (**right**), Scale bars, 4 mm; (**B**,**C**) Frequency of RORγt^+^ ILC3 cells and representative FACS plots of RORγt^+^ and GATA3^+^ cells among Lin^−^IL-7Rα^+^ cells gated on live CD45^+^ cells isolated from the lamina propria (**right**). Each symbol represents an individual mouse; mean ± SEM. * *p* < 0.05 (two-tailed unpaired Student’s *t*-test), ** *p* < 0.01 (one-way ANOVA).

**Figure 4 cells-11-01970-f004:**
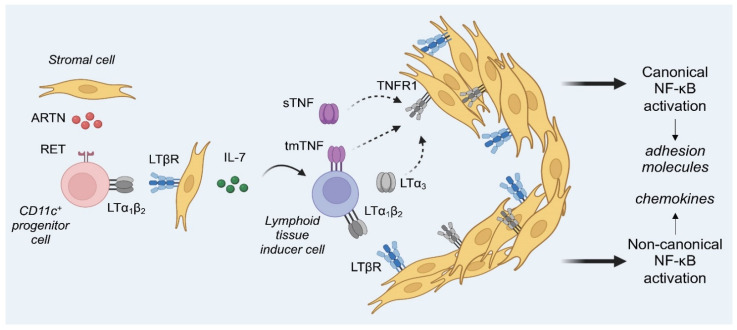
A model of a stepwise contribution of LTα and TNF in Peyer’s patch development. Stromal cells produce ARTN which binds to RET receptor on a CD11c^+^ hematopoietic progenitor cell. RET activation may lead to the induction of LTα_1_β_2_ on CD11c^+^ cells. CD11c^+^ cells then interact with LTβR^+^ stromal cells which produce IL-7 for the survival of LTi cells. LTi cells start to express LTα_1_β_2_, which interacts with LTβR on stromal cells leading to the activation of the non-canonical NF-κB pathway. LTi cells may also express tmTNF, sTNF, and LTα_3_ that bind to TNFR1 on a yet unknown cell type, presumably on stromal cells, leading to the activation of canonical NF-κB pathway. Both non-canonical and canonical NF-κB pathways synergize to produce chemokines and adhesion molecules that attract and retain lymphocytes in PP. Created with BioRender.com.

## Data Availability

Data supporting the reported results are available on request from the corresponding author.

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
