# Peer review of "LTα, TNF, and ILC3 in Peyer’s Patch Organogenesis"

_cells, 2022, doi:10.3390/cells11121970_

Round 1

Reviewer 1 Report

In this communication article, Dr. Nedospasov1 and colleagues to  clarify the controversial  contribution of  genes of the TNF/LT locus  in lymphoid tissue organogenesis. They claim that  their results  present new findings and discuss the distinct contribution of TNF produced by ILC3 cells to   Peyer Patches organogenesis._

 Comments:

 Line 144, the authors hypothesize that lack of PPT formation in Lta and Ltb ko mice could be acted by the retention of neo cassette that could influence closed genes expression such as Tnf encoding gene. Is there any report in the literature supporting this hypothesis or showing whether neighboring targeted gene are effect by neo cassette persistence?

Fig.1. Control Ltafl/fl is missing. This will help show that PP number is comparable to the heterozygote Lta+/DLtb +/Dmice.

The results contain text that is more appropriate for the discussion (description of Fig.1A).

Figure 1B: it is not clear why genetic modifier (neo cassette) do not affect PP development phenotype but does in the case of Lta and b. 

Figure 2 B. the difference observed could also explained by the reduction of ILC3 and CD4+ T cell number. Did the author check whether these cells affected in their mice?

The sequence in presenting data of Figure 2 should be reorganized.

The data could strengthen  by demonstrating that the gain of function using ILC3 TNF competent cells restores PP generation?

Fig 3: Not clear and no demonstration how PP are maintained in the intestinal lamina propria of 

Lta D/D mice. 

Author Response

The authors thank the reviewer for making very valuable comments. The point-by-point responses are below:

  1. Line 144, the authors hypothesize that lack of PPT formation in Ltaand Ltb ko mice could be acted by the retention of neo cassette that could influence closed genes expression such as Tnf encoding gene. Is there any report in the literature supporting this hypothesis or showing whether neighboring targeted gene are effect by neo cassette persistence?

We appreciate the reviewer’s comment and added the appropriate references to the text.

  1. Fig.1. Control Ltafl/flis missing. This will help show that PP number is comparable to the heterozygote Lta+/DLtb +/Dmice.

We added control Ltafl/fl mice in Figure 1A.

  1. The results contain text that is more appropriate for the discussion (description of Fig.1A).

We have modified the text accordingly.

  1. Figure 1B: it is not clear why genetic modifier (neo cassette) do not affect PP development phenotype but does in the case of Lta and b

We specified that TnfD/D mice do not harbor neo cassette in the description of Figure 1B and added a Supplementary Figure with the comparison of the designs for neo-free and conventional knockout mice.

  1. Figure 2 B. the difference observed could also explained by the reduction of ILC3 and CD4+ T cell number. Did the author check whether these cells affected in their mice?

Indeed, TNF ablation might affect CD4+ T cell and ILC3 numbers. However, ILC3 are crucial for the PP development (Eberl et al., Nat Immun, 2004), but T cell-deficient mice still develop PP anlagen.  We have not seen differences in T cell numbers in TNF KO mice when compared to wild-type littermate control mice. At the same time, ILC3 numbers in TNF KO were actually increased (Figure 3C). Thus, we believe that the observed phenotype is not due to reduced numbers of aforementioned cell subsets. Text was modified accordingly.

  1. The sequence in presenting data of Figure 2 should be reorganized.

We reorganized Figure 2 according to the reviewer’s suggestion.

  1. The data could strengthen  by demonstrating that the gain of function using ILC3 TNF competent cells restores PP generation?

We appreciate the reviewer’s comment. Indeed, there are protocols for transfer of purified ILC3 (Guo et al., Methods Mol Biol, 2016). However, lymphoid tissue inducer cells that are implicated in the generation of PP anlagen have a unique and dynamic phenotype during embryogenesis (van de Pavert, Biomed J, 2021). Thus, we speculate that embryonic LTi may have distinct features when compared to adult LTi-like cells.

  1. Fig 3: Not clear and no demonstration how PP are maintained in the intestinal lamina propria of Lta D/D

We would like to clarify that LtaD/D mice do not develop PP (Figure 1A). In control mice Peyer’s patches were excised before analysis of the intestinal lamina propria.

Reviewer 2 Report

The authors touch a very important topic that relates with the generation of secondary lymphoid structures in the gut, place in which tolerance/immune response to pathogens is key feature for survival. The manuscript is well written and methodology sufficiently detailed. Conclusions are properly sustained by the results. I would just suggest authors to speculate a bit on the fact that cecal PP are generated independently of the TNF/LT axis, in this case which would be the mechanisms involved.

Author Response

We thank the reviewer for the positive assessment of our manuscript. 

We appreciate the reviewer’s comment. Cecal patch development is critically dependent on the activity of NIK kinase which mediates LTbR signaling (Masahata et al., Nat Comm, 2014) and in utero LTbR-Ig treatment blocks cecal patch development (Donaldson et al., J Virol, 2015).

Reviewer 3 Report

Title: LTα, TNF and ILC3 in Peyer’s patch organogenesis 

Manuscript #1739417

General Remarks

This is an interesting study reanalysing the roles of TNF and LTα in of Peyer's patch development. I have only minor suggestions to improve the impact of the study.

Specific Remarks

Line 31 - Could cite Etemadi et al 2013 FEBS J for statement “sTNF and 30 sLTα3, respectively, that bind to and signal through both TNFR1 and TNFR2”

Fig. 1 while anyone can look up and find the organisation of the TNF locus I think it would help the average reader if the locus was represented schematically. Extra marks for showing the neo locus in strains of knock-out mice used in older studies.

Line 149 - Fig. 1a doesn't have Ltbr mice.

Fig. 2a font size needs to be larger, almost unreadable.

Fig. 3a images of PP are too small need some magnified images, also no scale bar.

Line 220 -  “development of the development” is incorrect.

Line 232 - ”paper” is superfluous.

Author Response

We thank the reviewer for making very valuable comments. The point-by-point responses are below:

  1. Line 31 - Could cite Etemadi et al 2013 FEBS J for statement “sTNF and 30 sLTα3, respectively, that bind to and signal through both TNFR1 and TNFR2”

We added the reference accordingly.

  1. Fig. 1 while anyone can look up and find the organisation of the TNF locus I think it would help the average reader if the locus was represented schematically. Extra marks for showing the neo locus in strains of knock-out mice used in older studies.

We thank the reviewer for this suggestion. We added a Supplementary Figure clarifying the design for different mouse strains targeting TNF/LT locus.

  1. Line 149 - Fig. 1a doesn't have Ltbr mice.

Corrected.

  1. Fig. 2a font size needs to be larger, almost unreadable.

We agree and therefore scaled up the font size for Figure 2B.

  1. Fig. 3a images of PP are too small need some magnified images, also no scale bar.

We added magnified images and a scale bar.

  1. Line 220 -  “development of the development” is incorrect.

Corrected.

  1. Line 232 - ”paper” is superfluous.

Corrected.
